# ZOOMING INTO COMICS: REGION-AWARE RL IMPROVES FINE-GRAINED COMIC UNDERSTANDING IN VISION-LANGUAGE MODELS

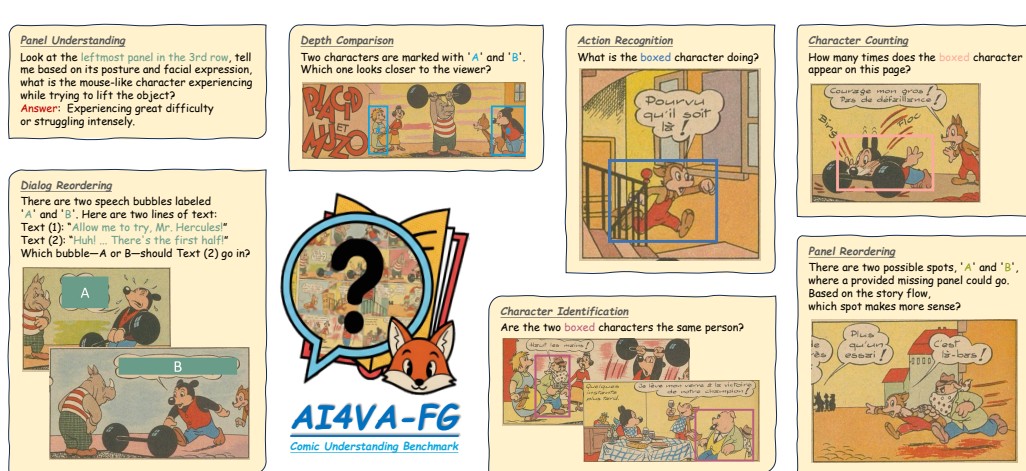

Figure 1: **Benchmark Overview.** (a) We introduce AI4VA-FG, a comic-centric benchmark featuring full-page, long-form narratives designed to challenge modern Vision-Language Models (VLMs). (b) The benchmark includes seven distinct tasks that evaluate a range of capabilities, from foundational recognition and detection to high-level plot and character understanding. (c) Our analysis shows that state-of-the-art VLMs struggle, but their performance is significantly boosted by post-training techniques, especially our proposed Region-Aware Reinforcement Learning.

## ABSTRACT

Complex visual narratives, such as comics, present a significant challenge to Vision-Language Models (VLMs). Despite excelling on natural images, VLMs often struggle with stylized line art, onomatopoeia, and densely packed multi-panel layouts. To address this gap, we introduce **AI4VA-FG**, the first fine-grained and comprehensive benchmark for VLM-based comic understanding. It spans tasks from foundational recognition and detection to high-level character reasoning and narrative construction, supported by dense annotations for characters, poses, and depth. Beyond that, we evaluate state-of-the-art proprietary models, including GPT-4o and Gemini-2.5, and open-source models such as Qwen2.5-VL, revealing substantial performance deficits across core tasks of our benchmarks and underscoring that comic understanding remains unsolved. To enhance VLMs' capabilities in this domain, we systematically investigate post-training strategies, including supervised fine-tuning on solutions (SFT-S), supervised fine-tuning on reasoning trajectories (SFT-R), and reinforcement learning (RL). Beyond that, inspired by the emerging "Thinking with Images" paradigm, we propose **Region-Aware Reinforcement Learning (RARL)** for VLMs, which trains models to dynamically attend to relevant regions through zoom-in operations. We observe that when applied to the Qwen2.5-VL model, RL and RARL yield significant gains in low-level entity recognition and high-level storyline ordering, paving the way for more accurate and efficient VLM applications in the comics domain.[1]

---

[1]The benchmark and evaluation scripts will be released publicly upon publication.

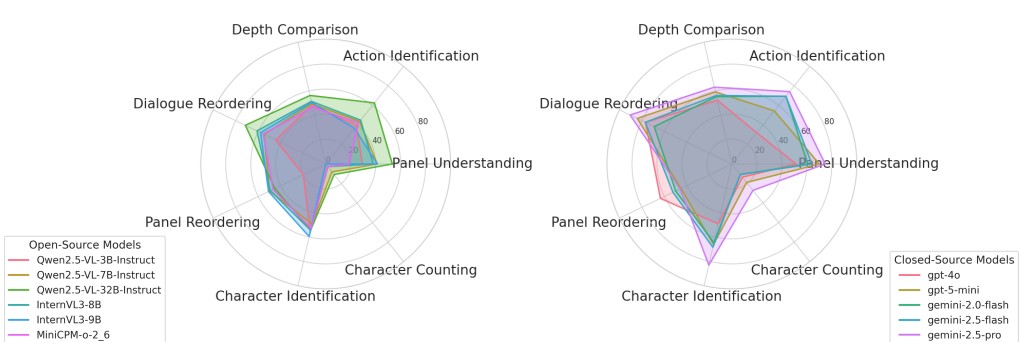

Figure 2: **Performance of state-of-the-art open-source and proprietary models on AI4VA-FG.** Although proprietary models achieve strong accuracy on most tasks, their performance remains inconsistent across the benchmark. Open-source models, in contrast, exhibit a 10–30 percentage point deficit, with the most pronounced weaknesses in depth perception, character tracking, and narrative construction.

## 1 INTRODUCTION

Vision-Language Models (VLMs) have demonstrated remarkable success on basic visual tasks involving single, static images like captioning and visual question answering. However, their capabilities are largely untested on complex visual narratives, which demand reasoning across long-form, full-page formats. Comics, in particular, present a significant challenge by requiring models to understand both the intricate details within a dense page and the long-range dependencies of a discrete storyline. To address this gap, we introduce **AI4VA-FG**, a benchmark specifically designed to evaluate and advance VLM capabilities for comic understanding, encompassing entity recognition, spatial perception, character tracking, and narrative construction.

Comics combine stylized line art, onomatopoeia, and densely packed speech balloons into multi-panel layouts that demand both fine-grained perception and higher-level reasoning (Vivoli et al., 2024b). To assess how well current VLMs handle this challenging domain, we construct **AI4VA-FG** (AI4VA Fine-Grained), the first comprehensive benchmark specifically designed for fine-grained comic understanding. AI4VA-FG spans low-level recognition, mid-level entity identification across panels, and high-level storyline reordering, with all questions enriched by dense panel- and character-level bounding box annotations. We benchmarked both proprietary systems, including GPT-4o (Hurst et al., 2024) and Gemini-2.5 (Team, 2025a), and open-source model series, including Qwen2.5-VL (Team, 2025b) and InternVL3 (Zhu et al., 2025), on AI4VA-FG. While the proprietary models achieve solid accuracy on most tasks, their performance is not uniformly strong across all tasks; meanwhile, open-source models lag behind by 10–30 percentage points, with the largest deficits observed in depth perception, character tracking, and storyline narrative construction.

These limitations can be attributed to three primary factors. First, there exists a substantial domain shift: the AI4VA corpus is composed of mid-twentieth-century Franco-Belgian comics differing markedly from the natural images and contemporary digital art that predominate in VLM benchmarking. Second, the available supervision for comics is insufficiently fine-grained: existing comic datasets typically provide annotations at the level of entity masks and relationships, but rarely capture details such as depth layering and subtle pose variation. Third, current VLMs lack explicit mechanisms for selective visual focus when confronted with high-resolution comic pages: a single page often contains over ten densely illustrated panels; encoding the entire page as a monolithic input both strains the context window and disperses attention, whereas human readers naturally adopt a zoom-in strategy to attend to relevant regions prior to higher-level reasoning.

We conducted a systematic investigation of two principal post-training paradigms for VLMs, supervised fine-tuning (SFT) and reward-based reinforcement learning (RL), together with two sub-variants of each, using the training split of AI4VA-FG. SFT raised most accuracy by limited percentage but left depth comparison almost unchanged, and can improve dialogue reordering only when supervised with high-quality reasoning trajectories. RL exceeds SFT on most tasks but can

only slightly improve reordering tasks. In addition, RL demonstrates greater cross-task generalization than SFT: training on a subset of challenging comic reordering tasks improves performance on easier recognition tasks.

Furthermore, motivated by the observation that page size constrains models' comprehension of individual panels, and inspired by the "Thinking with Images" approach of OpenAI-o3 (Hurst et al., 2024), we introduce Region-Aware Reinforcement Learning (Region-Aware RL, or RARL). During reasoning, the VLM may issue a zoom-in command with a bounding-box argument to retrieve the selected patch and incorporate it into its context before continuing to reason. RARL is trained in two stages: (i) a warm-start phase that teaches basic tool-calling behavior, and (ii) a hierarchical RL phase that rewards accurate zoom-in operations and, in particular, tool usages that contribute to correct final answers. Applied to Qwen2.5-VL-7B-Instruct, RARL increases depth-comparison accuracy by 7.3% and action-recognition accuracy by 32.7%, closing over half the gap to Gemini-2.5-Pro. The main contributions are summarized as follows:

- We introduce **AI4VA-FG**, the first fine-grained benchmark for comic understanding with VLMs, featuring both low-level recognition tasks and high-level reasoning tasks with dense annotations for character, actions, depth, etc.
- We benchmark a range of state-of-the-art vision-language models on AI4VA-FG and provide a detailed analysis of their performance and common failure cases.
- We evaluate post-training methods including SFT and RL on improving model performance across comic understanding tasks.
- We propose **Region-Aware Reinforcement Learning (RARL)**, a reinforcement learning framework inspired by the emerging "Thinking with Images" paradigm, which learns both *where* and *when* to zoom for more effective visual reasoning in comics.

## 2 RELATED WORK

**Benchmarks for Comics**. Recently, comic benchmarks have shifted from fundamental tasks—such as object detection, speaker identification, and reading order detection (Vivoli et al., 2024a)—targeting specific models to more comprehensive tasks tailored for VLMs. *MangaUB* (Ikuta et al., 2024) and *MangaVQA* (Baek et al., 2025), targeting manga—the Japanese comic art form—assess both panel-level recognition and multi-panel comprehension through manually curated question–answer pairs spanning diverse narrative scenarios. In contrast, *ComicsPAP* (Vivoli et al., 2025) and *StripCipher* (Wang et al., 2025) emphasize multi-panel reasoning tasks such as panel prediction and reordering but lack fine-grained entity tracking, highlighting the performance gap between current VLMs and human capabilities in understanding sequential narratives. While some of these benchmarks have explored SFT, none have applied RL.

Table 1: **Features and statistical information of AI4VA-FG and prior related benchmarks**. #QA refer to the number of question-answer pairs. "Public" indicates whether the images are sourced from the public domain, ensuring that the benchmark can be distributed without legal restrictions. AI4VA-FG is, to our knowledge, the only publicly available benchmark for comic understanding that has been systematically evaluated using both SFT and RL.

| Benchmark | Task Categories | #QA | SFT | RL | Public |
|---|---|---|---|---|---|
| *MangaUB* | Recognition, Comprehension, Reordering | 6,585 | × | × | × |
| *StripCipher* | Comprehension, Reordering | 2,170 | ✓ | × | × |
| *ComicsPAP* | Reordering | 103,933 | ✓ | × | ✓ |
| *MangaVQA* | Recognition, Comprehension | 40,363 | ✓ | × | × |
| ***AI4VA-FG*** | Recognition, Comprehension, Reordering | 16,264 | ✓ | ✓ | ✓ |

**Post-Training of VLMs**. The post-training phase is essential for enhancing pretrained large language models' capabilities for real-world deployment. This stage predominantly encompasses two methodologies: supervised fine-tuning (SFT) and reinforcement learning (RL) (Chu et al., 2025). Recently, DeepSeek-R1 (DeepSeek-AI, 2025) demonstrated substantial improvements in text-based reasoning through RL with rule-based rewards, and subsequent studies (Liu et al., 2025) have further validated the effectiveness of pure RL in enhancing visual reasoning capabilities. Notably,

RL demonstrates substantially stronger generalization capabilities than SFT when handling out-of-distribution (OOD) multimodal tasks (Chen et al., 2025; Chu et al., 2025; Rajani et al., 2025).

**Thinking with Images**. To emulate the human ability to process complex visual information through selective attention, VLMs can learn to dynamically identify salient regions within an image and adaptively "zoom in" to form a visual chain of thought (CoT) (Shao et al., 2024a). "Thinking with Images" is a recent visual reasoning paradigm in which image manipulation tools—such as zoom-in or cropping—are used to transform the input image, enabling VLMs to better comprehend and reason about visual content. Recently, *DeepEyes* (Zheng et al., 2025) adopts an end-to-end RL paradigm (without SFT cold start) to incentivize tool-assisted visual reasoning; it does not incorporate explicit guidance for tool-usage accuracy, which limits its training efficiency. Meanwhile, some studies (Su et al., 2025; Zhang et al., 2025) adopt a two-stage post-training approach: firstly, grounding capabilities are established through instruction tuning; subsequently, visual reasoning is enhanced via RL. While these methods predicting variable-size bounding boxes for zoom-in, (Kumar et al., 2025) predicts points and does fixed-size crops to reduce end-to-end RL training costs.

## 3 THE AI4VA-FG BENCHMARK

Considering the aforementioned limitations of existing benchmarks, we introduce a new comics benchmark, AI4VA-FG (AI4VA Fine-Grained), the first benchmark specifically designed for both low-level and high-level tasks in comics, incorporating entity-level recognition as well as temporal reasoning questions.

We develop our benchmark based on the AI4VA dataset (Grönquist et al., 2024), which offers a rich and diverse collection of comic-style imagery sourced from two mid-twentieth-century Franco-Belgian comics series, *Placid et Muzo* and *Yves le loup – Bandes Dessinées*, whose faded colors, halftone shading, and hand-lettered typography differ markedly from the natural images and contemporary digital art that predominate in VLM benchmarking. The dataset offers dense annotations of semantic segmentation, ordinal depth, and visual saliency for each comic page.

We initially employ a scripted pipeline to generate questions from the segmentation labels in AI4VA. These automatically generated questions are then refined through a manual filtering process to ensure clarity and semantic alignment with the visual content. All VQA instances are equipped with bounding box annotations, a design choice that ensures suitability for agentic RL training.

### 3.1 TASK DEFINITIONS

Based on their contextual scope, all questions can be categorized into two types: **single-panel** and **multi-panel**. Single-panel questions are grounded within the content of a single comic panel, while multi-panel questions require reasoning across multiple panels within a page. Each of these two types is further divided into two subcategories: **recognition** and **understanding** tasks. Recognition tasks focus on extracting explicitly presented information, while understanding tasks involve inferring implicit or internal information embedded in the visual and narrative context. Our benchmark includes seven VQA tasks, generally arranged in order of increasing difficulty.

The single-panel tasks are: **Panel Understanding**, which evaluates the model's ability to locate and interpret a specific panel; **Action Recognition**, which measures its capacity to identify a marked character's posture or action; and **Depth Comparison**, which tests spatial reasoning by comparing the relative depth of characters. The multi-panel tasks target higher-level narrative reasoning and character tracking. **Dialog Reordering** assesses narrative coherence by reconstructing the correct order of shuffled dialog balloons, while **Panel Reordering** evaluates story progression and visual continuity by placing a missing panel in sequence. **Character Identification** probes whether two characters across panels are the same entity, while **Character Counting** measures how accurately the model tracks a character's appearances across a page.

As illustrated in Fig. 1, for Panel Understanding tasks the panel positions are provided only as textual descriptions rather than visual markings, requiring models to perform grounding solely from the text prompts; in contrast, other tasks explicitly mark relevant characters or panels in the image, making grounding easier. All tasks are divided into standard training, validation, and test splits. With the

exception of the Dialog and Panel Reordering tasks, which lack defined relevant panels, all other tasks' VQA triples are annotated with bounding boxes of the corresponding panels or characters.

Table 2: **Summary of benchmark tasks and their associated statistics**. #Ch. denotes the number of answer choices per multi-choice question, and #QA refers to the total number of VQAs for each task. The Character Counting task requires numerical answers, while the Panel Understanding task requires open-text responses that rely on LLM-as-a-Judge (Zheng et al., 2023) for verification; all other tasks utilize multiple-choice answers.

| Category | Task | Type | #Ch. | #QA |
|---|---|---|---|---|
| Single-Panel Understanding | Panel Understanding | open-ended | / | 7902 |
| Single-Panel Recognition | Action Recognition | multi-choice | 4 | 1669 |
| | Depth Comparison | multi-choice | 2 | 1125 |
| Multi-Panel Understanding | Dialog Reordering | multi-choice | 2 | 1364 |
| | Panel Reordering | multi-choice | 2 | 1392 |
| Multi-Panel Recognition | Character Identification | multi-choice | 4 | 2368 |
| | Character Counting | numerical | / | 444 |
| **Total** | | | | **16264** |

## 3.2 Performance Analysis

We evaluate selected VLMs on AI4VA-FG's test split, with results reported in Tab. 3.2. Overall, proprietary models outperform the open-source counterparts, and Gemini-2.5-pro achieves the highest accuracy in 6 out of 7 tasks. Interestingly, open-source models that rank higher on general VLM leaderboards still fail to surpass commercial models on our benchmark. Despite these advances, a substantial gap persists between current VLMs and human-level comic understanding, primarily due to their pronounced limitations in spatial perception, character tracking, and multi-panel narrative construction.

Table 3: **Evaluation results of state-of-the-art VLMs on AI4VA-FG**. While proprietary models generally outperform open-source counterparts, a performance gap remains to human-level understanding. Notably, most models perform close to random chance on Depth Comparison, Panel Reordering, and Character Counting. Best and second-best performance values (that exceed random-guess accuracy) are indicated using **bold** and underlined formatting, respectively.

| Model | Panel Understanding | Action Recognition | Depth Comparison | Dialog Reordering | Panel Reordering | Character Identification | Character Counting |
|---|---|---|---|---|---|---|---|
| Random | / | 25.00 | 50.00 | 50.00 | 50.00 | 50.00 | / |
| Qwen2.5-VL-3B-Instruct | 29.33 | 39.46 | 48.54 | 44.44 | 19.84 | 55.88 | 2.70 |
| Qwen2.5-VL-7B-Instruct | 41.33 | 43.54 | 49.51 | 54.76 | 50.00 | 49.26 | 8.11 |
| Qwen2.5-VL-32B-Instruct | 53.33 | 62.59 | 56.31 | 71.43 | 45.24 | 54.41 | 10.81 |
| InternVL3-8B | 38.00 | 44.90 | 51.46 | 61.11 | 49.21 | 54.41 | 2.70 |
| InternVL3-9B | 41.33 | 36.73 | 50.49 | 57.94 | 50.79 | 59.56 | 0.00 |
| MiniCPM-o-2.6 (8B) | 18.67 | 42.18 | 46.60 | 55.56 | 46.83 | 52.94 | 2.70 |
| GPT-4o-2024-08-06 | 51.33 | 31.97 | 52.43 | 76.80 | **63.49** | 49.22 | 13.51 |
| GPT-5-mini-2025-08-07 | 70.00 | 54.42 | 59.22 | 84.13 | 45.24 | 66.18 | 18.92 |
| Gemini-2.0-flash | 60.67[1] | 69.39 | 55.34 | 69.05 | 50.00 | 64.71 | 10.81 |
| Gemini-2.5-flash | 64.00[1] | 69.39 | 56.31 | 76.98 | 52.38 | 68.38 | 10.81 |
| Gemini-2.5-pro | **74.67**[1] | **74.15** | **63.11** | **90.48** | 46.03 | **83.09** | **27.03** |

**Depth Perception**. The majority of evaluated models exhibit poor spatial perception when processing comic images, performing near random when comparing entity depth. Compared to depth perception in real-world images (Fu et al., 2024), GPT-4o's performance on comics is notably more random, suggesting that the stylistic nature of comic drawings introduces additional challenges for spatial reasoning. **Narrative Reordering**. While Gemini-2.5-flash and GPT-5-mini achieve promising performance on Dialog Reordering, their accuracy on Panel Reordering remains near random chance (50%). OCR-extracted dialog text is supplied in Dialog Reordering, aiding the model in reconstructing the narrative flow across adjacent panels. However, in the absence of such textual guidance, VLMs exhibit markedly constrained ability to compose coherent narratives solely from discrete sequential images. **Character Tracking**. Each image in AI4VA contains on average 13

---

[1]Gemini's high accuracy on Panel Understanding cannot be regarded as a fair measure, since both the questions and answers in this task were generated by Gemini.

panels, making it particularly challenging to track all appearances of a character across panels, especially when individual appearances are too small to be reliably identified. This difficulty accounts for the poor performance of all models on Character Counting.

Table 4: Accuracy (%) of selected models on AI4VA-FG with entire-page and individual-panel inputs, respectively. Zoom-in on relevant individual panels yields significantly improved accuracy.

| Model | Panel Understanding | Action Recognition | Depth Comparison | Character Identification |
|---|---|---|---|---|
| Qwen2.5-VL-7B | 41.33 | 43.54 | 49.51 | 49.26 |
| (zoom-in) | +17.34 58.67 | +12.24 55.78 | +4.83 54.34 | +3.68 52.94 |
| GPT-4o | 51.33 | 31.97 | 52.43 | 49.22 |
| (zoom-in) | +21.34 72.67 | +23.13 55.10 | +6.79 59.22 | +14.75 63.97 |
| Gemini-2.5-Flash | 64.00 | 69.39 | 56.31 | 68.38 |
| (zoom-in) | +16.00 80.00 | +6.12 75.51 | +22.33 78.64 | +6.62 75.00 |

**Does Zooming-In Improve Performance?** For humans, these tasks are challenging when viewing the entire page at a glance, but become considerably easier when focusing on the relevant panels. This motivates us to further compare the performance of entire-page versus individual-panel inputs for the single-panel tasks. We observe accuracy improvements across selected tasks when zooming solely into the relevant panels, likely due to the removal of unrelated content. These results highlight the importance of incorporating zoom-in mechanisms that enable models to focus on salient, detailed regions of a comic page, such as individual panels or characters.

## 4 METHODOLOGY

Noticing the large performance gap, we go beyond evaluating pre-trained VLMs and systematically study post-training strategies, including SFT and RL. For SFT, we consider two variants: fine-tuning on final solutions (**SFT-S**) and fine-tuning on filtered, verified synthetic reasoning trajectories (**SFT-R**). For RL, we evaluate both vanilla GRPO-based RL and our proposed agentic method, **Region-Aware Reinforcement Learning (RARL)**, which incentivizes zoom-in operations on relevant image regions with explicit guidance for tool-usage accuracy.

### 4.1 ENABLE "THINKING WITH IMAGES" VIA AGENTIC RL

We adopt a two-stage agentic RL framework: (1) a brief **warm-start phase** that leverages only basic tool usage rewards to establish tool-calling behavior, and (2) a main **RL training phase** that incorporates the complete reward structure to incentivize accurate and effective zoom-in actions. In contrast to other two-stage approaches that rely on a SFT cold-start, our warm-start phase remains entirely within the RL paradigm, differing solely in the reward configuration. As a result, it does not require any curated SFT datasets consisting of manually synthesized tool-calling trajectories.

**Reward Strategy.** In the context of VLMs, outcome-based rewards play a key role in steering models toward effective reasoning and decision-making. In our RL training phase, the total reward structure consists of three parts: an accuracy reward $R_{acc}$, a formatting reward $R_{format}$, and a tool usage reward $R_{tool}$. The accuracy reward measures whether the final answer is correct, while the formatting reward penalizes improperly structured outputs. The tool usage reward is given when an external tool is called correctly during the reasoning process. Formally, for a reasoning trajectory $\tau$, the total reward is:

$$R(\tau) = R_{format}(\tau) + R_{acc}(\tau) + R_{tool}(\tau), \tag{1}$$

The tool usage reward depends both on whether the external tool is invoked appropriately and on the accuracy of the tool's output relative to the given question. DeepEyes (Zheng et al., 2025) employs a strategy in which a constant tool usage bonus is added to the total reward only when the final answer is correct. In our setting, since each question includes a ground-truth region of interest (e.g., a character or panel region on the page), we propose a variant of the tool usage reward that more effectively encourages correct tool usage by explicitly measuring spatial accuracy:

$$R_{tool}(\tau) = (1 + \mathbb{I}_{R_{acc}(\tau)>0})(R_{tool\text{-}count}(\tau) + R_{tool\text{-}acc}(\tau)) \tag{2}$$

where $\mathbb{I}$ is an indicator function that activates an additional tool-usage bonus only when the final answer is correct, ensuring that the tool usage likely contributes to the outcome. $R_{\text{tool-count}}(\tau)$ denotes the reward component evaluating whether the number of zoom-in tool invocations in the reasoning trajectory matches the expected count, and $R_{\text{tool-acc}}(\tau)$ represents the accuracy-based bonus awarded for correct tool usages:

$$R_{\text{tool-acc}}(\tau) = \frac{1}{\sqrt{m}} \sum_{i=1}^{m} \text{IoU}(\tau_i) \tag{3}$$

Here, $\tau_i$ denotes the sub-trajectory corresponding to the $i$-th tool usage, and $m$ is the number of zoom-in tool invocations. For each predicted zoom-in bounding box, the Intersection over Union (IoU) is computed between the predicted box and its corresponding target region in the image. The accuracy bonuses $R_{\text{tool-acc}}(\tau_i)$ are summed to give higher rewards when multiple zoom-in operations are correctly performed. The normalization by $\sqrt{m}$ stabilizes the reward distribution when multiple bounding boxes are output for tasks such as Character Identification & Counting.

## 5 EXPERIMENTS

### 5.1 MAIN RESULTS

We train Qwen2.5-VL-7B-Instruct on $8 \times$ H800 (80G) GPUs, using LLaMA-Factory[2] and verl[3] frameworks for SFT and RL respectively. We adopt GRPO (Shao et al., 2024b) algorithm for RL training. The training details are provided in Appendix F.2.

Table 5: **Post-training methods' performance on AI4VA-FG tasks.** SFT-R outperforms SFT-S, and RL generally yields greater performance gains than SFT.

| Model | Panel Understanding | Action Recognition | Depth Comparison | Dialog Reordering | Panel Reordering | Character Identification | Character Counting |
|---|---|---|---|---|---|---|---|
| Qwen2.5-VL-7B | 41.33 | 43.54 | 49.51 | 54.76 | 50.00 | 49.26 | 8.11 |
| SFT-S (vanilla) | +8.67 50.00 | +26.53 70.07 | -3.88 45.63 | +0.80 55.56 | -2.98 47.02 | -0.04 49.22 | +10.81 **18.92** |
| SFT-R (reasoning) | +7.34 48.67 | +24.49 68.03 | -3.88 45.63 | +14.29 **69.05** | -0.79 49.21 | +14.71 63.97 | +2.70 10.81 |
| RL (vanilla) | +14.00 55.33 | +28.57 72.11 | +2.92 52.43 | +5.56 60.32 | -2.38 47.62 | +16.92 66.18 | -2.70 5.41 |
| **Region-Aware RL** | +10.00 51.33 | +32.64 **76.19** | +7.28 **57.28** | / | / | +22.06 **71.32** | -5.41 2.70 |

**SFT v.s. RL.** On most tasks, both SFT and RL yield significant improvements. Among the two SFT settings, SFT-R consistently outperforms SFT-S, demonstrating that CoT distillation enhances visual reasoning even for low-level recognition tasks. Furthermore, except for Dialogue Reordering and Character Counting, RL-finetuned models outperform SFT counterparts and reach or surpass certain proprietary models.

However, on the two reordering tasks, RL brings very limited improvement and lags far behind distillation with Gemini's CoT trajectories. This may be because the internal reordering capability of the 7B base model is substantially weaker than that of Gemini-2.5, and GRPO can only amplify existing abilities but struggles to create entirely new ones (Rajani et al., 2025). Specifically, we also apply RL on top of the distilled model, but its performance surprisingly gradually degrades: the model forgets the narrative reordering ability inherited from Gemini-2.5-flash and fails to acquire new effective reasoning patterns during RL training.

**Region-Aware RL**. RARL optimizes two objectives: grounding IoU and VQA accuracy. Since ground-truth bounding boxes are unavailable for the two reordering tasks, we fine-tune Qwen-2.5-VL-7B on the remaining five tasks using RARL. The results suggest that the model possesses a strong grounding ability, reaching nearly 80% IoU from 20% for zoom-in operations if trained on Action Recognition and Depth Comparison only. Grounding accuracy on the Panel Understanding task is lower, as the relevant panel is specified only in the prompt but not explicitly marked in the image, which increases the likelihood of errors when the model attempts to localize the correct panel.

---

[2]https://github.com/hiyouga/LLaMA-Factory
[3]https://github.com/volcengine/verl

Notably, when multiple zoom-in operations are performed, the second operation is less accurate than the first, suggesting that limited context length constrains the model's grounding accuracy.

Table 6: Zoom-in Statistics of models finetuned via RARL.

| Task | Panel Understanding | Action Recognition | Depth Comparison | Character Identification |
|------|---------------------|--------------------|------------------|-------------------------|
| Avg. #Toolcall | 1.01 | 1.10 | 0.93 | 1.88 |
| Avg. IoU | 0.565 | 0.862 | 0.847 | 0.835 (1$^{st}$: 0.842; 2$^{nd}$: 0.829) |

Tab. 5.1 demonstrates RARL outperforms vanilla RL and even surpasses Gemini-2.5-Flash on three recognition tasks, achieving performance comparable to the latter with manual cropping (see Tab. 4). Its weaker improvement in Panel Understanding may stem from imprecise panel localization due to implicit position prompts. Overall, the results of RARL highlight that a smaller model, when equipped with appropriate post-training strategies, can exceed the performance of a much larger model on specific tasks. It also underscores the potential of tool-augmented reasoning to enhance model performance in scenarios involving large and visually dense contexts.

Notably, RARL brings no improvement on Character Counting, as the model fails to acquire a "crop-all-panels" strategy through pure RL for exhaustively checking character occurrences. We attribute this failure to two factors: (i) limited training samples for this task when jointly trained with other tasks, and (ii) degraded grounding accuracy as context length increases—when more than two panels have already been cropped. Addressing these issues likely requires stronger supervision or curriculum strategies to stabilize sequential zooming and bounding-box prediction.

## 5.2 ABLATIONS

Table 7: **Cross-Task Generalization Performance.** "SFT-R (character)" refers to the model finetuned on Character Identification & Counting tasks via SFT-R, while "RL (reorder)" refers to the model finetuned on Dialog & Panel Reordering tasks via vanilla RL. RL demonstrates stronger in-domain cross-task generalization than SFT, particularly for recognition-oriented tasks.

| Model | Panel Understanding | Action Recognition | Depth Comparison | Dialog Reordering | Panel Reordering | Character Identification | Character Counting |
|-------|---------------------|--------------------|--------------------|-------------------|------------------|--------------------------|--------------------|
| Qwen2.5-VL-7B | 41.33 | 43.54 | 49.51 | 55.56 | 50.00 | 50.78 | 8.11 |
| SFT-R (action & depth) | -2.00 39.33 | 63.95 | 42.72 | -20.64 34.92 | -35.71 14.29 | -8.13 42.65 | -2.70 5.41 |
| SFT-R (reorder) | +10.00 **51.33** | +8.16 51.70 | -6.79 42.72 | 71.43 | 50.79 | +19.07 **69.85** | 8.11 |
| SFT-R (character) | +5.34 46.67 | 43.54 | -1.94 47.57 | -4.77 50.79 | -10.32 39.68 | 63.28 | 2.70 |
| **RL (action & depth)** | +0.67 42.00 | 76.19 | 58.25 | -2.39 53.17 | -0.79 49.21 | +10.25 **61.03** | +2.70 10.81 |
| **RL (reorder)** | -2.00 39.33 | +22.45 65.99 | -6.79 42.72 | 57.94 | 44.44 | +11.72 **62.50** | -2.70 5.41 |
| **RL (character)** | +2.00 43.33 | +21.09 **64.63** | 49.51 | -1.59 53.97 | +0.79 50.79 | 69.12 | 10.81 |

**In-Domain, Cross-Task Generalization.** To investigate the generalization ability of the post-training methods, we train the model on each of the 3 closed-ended categories and evaluate the finetuned model on the other two. We observe that RL exhibits certain in-domain generalizability as (Chu et al., 2025; Chen et al., 2025) argues. For instance, Action Recognition benefits most from RL: training on multi-panel tasks also brings improvements on Action Recognition without obvious degradation on other tasks.

To assess whether this gain results merely from additional training, we performed an ablation study in which the model was trained with RL using only format rewards and random accuracy rewards. No appreciable performance gains were observed, suggesting that the improvement is driven by the model's ability to leverage meaningful in-domain reward signals rather than by increased training alone. Nevertheless, no performance gains are observed on Depth Comparison or Panel Reordering, supporting the argument that GRPO primarily amplifies existing capabilities already acquired during pre-training rather than enabling new forms of reasoning (Rajani et al., 2025).

Interestingly, SFT does not solely overfit to the supervised task, but can also demonstrate cross-task generalization: fine-tuning the base model with reasoning trajectories generated by Gemini-2.5-flash only for the reordering tasks results in superior performance (69.85%) also on Character Identification, even surpassing the model trained on this task via RL (69.12%). While Gemini-2.5-flash achieves high accuracy and generates high-quality reasoning on the Dialogue Reordering task,

this capability can be transferred to other tasks via distillation. Nevertheless, the extent of SFT in-domain generalization is inherently constrained by the amount and quality of the supervision data.

**Cross-Domain Generalization.** We further evaluate the fine-tuned model on MangaVQA (Baek et al., 2025), a benchmark for manga that is closely related to but distinct from comics. Neither SFT nor RL demonstrates cross-domain generalization on this dataset, while SFT leads to significant degradation. This limitation can be attributed to pronounced domain shifts, including differences in artistic style, panel layout, and text density between Western comics and Japanese manga.

Table 8: **Performance of finetuned models on MangaVQA-test and MMLU-val**. RL exhibits minimal degradation on general-domain tasks, whereas SFT incurs higher cross-domain drops.

| Method | MangaVQA-test | MMLU-val |
|---|---|---|
| Original | 22.78 | 53.78 |
| **SFT-R** | 10.58 | 51.11 |
| **RL** | 22.00 | 53.56 |

In addition, we evaluate the general reasoning capabilities of the RL-trained vision model using the widely adopted MMMU Yue et al. (2024) benchmark's validation split. After fine-tuning on six closed-ended tasks, the RL-optimized model exhibits minimal performance degradation on general-domain tasks in MMMU, whereas the SFT-fine-tuned model experiences greater performance drops, indicating that RL may confer better cross-domain robustness than SFT.

**Reward Strategy.** While DeepEyes (Zheng et al., 2025) argues that end-to-end RL alone is sufficient to enable tool usage, concurrent works (Su et al., 2025; Zhang et al., 2025) highlight the necessity of an SFT cold-start phase for achieving stable and effective learning. In our experiments, we found that fully end-to-end RL often leads to reward hacking during the early stages of training—where the model exploits only easily achievable components of the reward function, such as format correctness, without improving on tool use or reasoning. To address this, we adopt a two-phase RL strategy, in which the warm-start phase employs a simplified reward formulation to guide the model toward meaningful tool-using behaviors before full reward optimization begins.

We further experimented by removing the constant coefficient in the reward rule Equation 2, making the tool usage reward $R_{\text{tool}}$ conditional on the correctness of the final answer. This modification led to slow convergence in both tool accuracy and overall task accuracy: the model performs poorly on both targets at the early stage so that conditioning one on the other can greatly slow the learning of tool calling (see Appendix F.3). The results suggest that tool usage should consistently be rewarded whenever zoom-in operations are known to be beneficial, in order to enable more efficient tool learning.

## 6 CONCLUSION AND FUTURE WORK

We presented AI4VA-FG, the first fine-grained and comprehensive benchmark for comic understanding with VLMs, spanning both low-level recognition and high-level reasoning tasks with dense annotations. Through extensive evaluation of state-of-the-art models, we revealed persistent weaknesses in spatial perception, character tracking, and multi-panel narrative construction, underscoring the gap between open-source and proprietary systems. To mitigate these challenges, we examined post-training strategies, showing that both SFT and RL can yield cross-task generalization, while our proposed Region-Aware RL leverages zoom-in operations to improve grounding and narrative reasoning. Together, our benchmark and methods establish a foundation for advancing multimodal reasoning in the domain of comics.

In future research, more comic datasets can be transformed into comprehensive benchmarks for VLMs to support large-scale training. Beyond recognition and reordering tasks, incorporating basic tasks such as speaker identification would enable a more complete evaluation of comic understanding. Furthermore, the focus can be extended from understanding to generation; for instance, our VQA benchmark could be repurposed to assess the quality of comic storytelling by replacing image inputs with their corresponding generated captions. This would allow systematic evaluation of models' narrative generation capabilities, bridging the gap between visual comprehension and multimodal creative reasoning. The pipeline should also be enhanced in long-context settings where large number of zoom-in operations are required, to enable better performance on those tasks requires all panels such as character counting or page-level captioning.

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

## A    ETHICS STATEMENT

This work follows the ICLR Code of Ethics. No human subjects or animal experiments were involved. All datasets, including **AI4VA-FG**, were collected in accordance with usage guidelines and without violating privacy. We took care to minimize potential biases and discriminatory outcomes. No personally identifiable information was used, and no experiments were conducted that could raise privacy or security concerns. We are committed to ensuring transparency and integrity throughout the research process.

## B    REPRODUCIBILITY STATEMENT

We have taken extensive steps to ensure reproducibility. All codes and datasets will be released in an anonymous repository AI4VA-FG to facilitate replication and verification. The experimental setup, including training steps, model configurations, and hardware specifications, is described in detail in the paper. We also provide a complete description of **Region-Aware RL** to support faithful reproduction of our results. These measures aim to enable the community to reproduce our work and build upon it.

## C    THE USE OF LARGE LANGUAGE MODELS (LLMS)

Large Language Models (LLMs) were employed in a limited capacity during the preparation of this manuscript. Specifically, ChatGPT and Gemini were used solely for language refinement, such as polishing phrases and improving readability. All ideas, analyses, and conclusions presented in this paper are entirely the authors' own.

# D  BENCHMARK OVERVIEW

## D.1  DATASET CONSTRUCTION

**Open-Ended Task**. The process of constructing the Panel Understanding task consists of four steps: (1) crop each comic page into individual panels and use Gemini-2.5-flash to generate captions for all panels; (2) based on the panel images and captions, prompt Gemini-2.5-flash to generate several pairs of original questions and answers; (3) We employ a specialized panel-ordering model to index all panels within each page and generate textual descriptions of their positions, followed by manual verification to ensure positional accuracy; (4) concatenate the positional descriptions with the original questions to form the final VQA triplets *(comic page image, question, answer)*. Since the ground-truth answers for the Panel Understanding task are open-text, an LLM-as-a-Judge (Zheng et al., 2023) approach is employed to verify model responses.

**Closed-Ended Tasks**. For all other closed-ended tasks, we develop a pipeline framework to transform segmentation annotations into a QA format compatible with LLMs. Using this framework, we convert AI4VA's segmentation annotations into six tasks comprising roughly 8k triples, and we will release the pipeline to enable the generation of additional VQAs for other comic datasets when large-scale training is required.

## D.2  BENCHMARK STATISTICS

Statistics of AI4VA-FG benchmark: train-validation-test splitting, task categories, answer types, panel position prompt types, and whether tasks are single- or multi-panel.

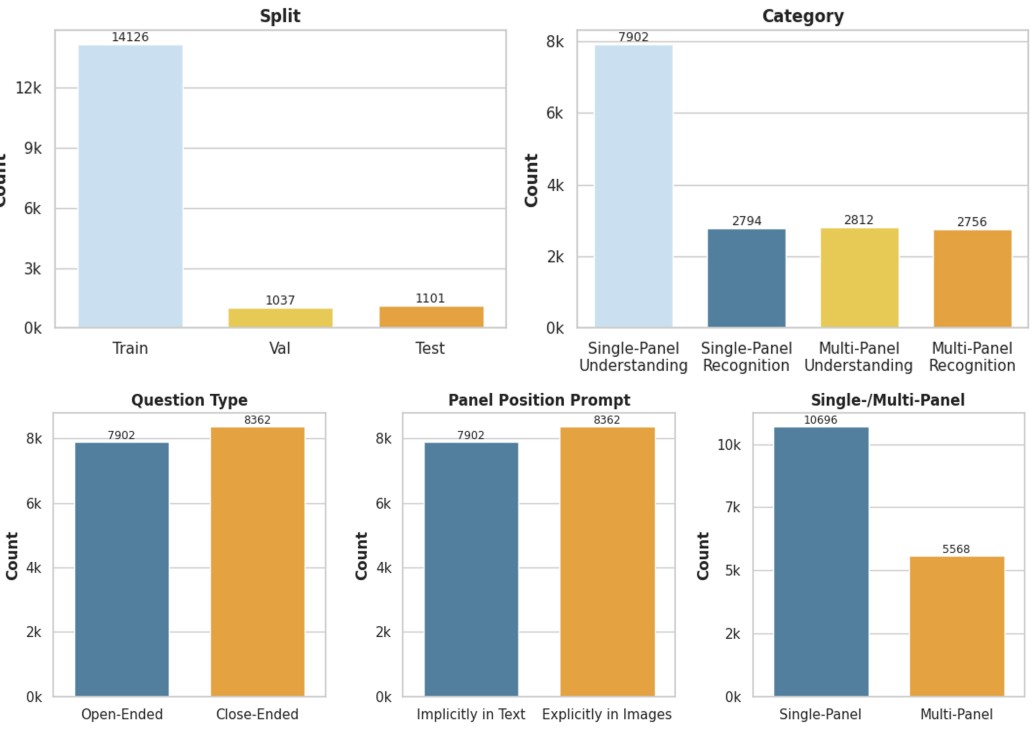

Figure 3: Statistics of our AI4VA-FG benchmark.

# E MORE CASES

## E.1 COMMON FAILURE CASES ON COMIC UNDERSTANDING

**Depth Comparison**

- **Question**: Two characters are marked with 'A' and 'B'. Which one looks closer to the viewer?
- **Ground-Truth Answer**: B
- **Model**: Qwen2.5-VL-32B-Instruct
- **Predicted Answer**: A
- **Failure Analysis**: Incorrect spatial perception in comics. The model misjudges depth by assuming larger characters are always closer, ignoring intrinsic size differences (e.g., character A being naturally larger than B).
- **Full Predicted Answer**:

```
"To determine which character, A or B, looks closer to the viewer, we need to analyze the visual cues in the
image:\n\n1. **Size**: Characters that appear larger are generally perceived as being closer to the
viewer.\n2. **Position**: Characters positioned lower in the frame tend to be closer, as they are often
depicted as being in the foreground.\n3. **Details**: More detailed features (like facial expressions,
clothing details) suggest proximity.\n4. **Perspective**: The angle and perspective can indicate depth, with
objects closer to the viewer appearing more prominent.\n\n### Analysis:\n- **Character A**: In several
panels, Character A appears larger and more detailed compared to Character B. Additionally, Character A is
often positioned lower in the frame, suggesting a foreground placement.\n- **Character B**: While Character B
is also prominent, it does not consistently appear larger or more detailed than Character A. In many panels,
Character B is slightly smaller or less detailed.\n\n### Conclusion:\nBased on the size, position, and
detail, Character A appears closer to the viewer.\n\n<answer>A</answer>"
```

**Character Identification**

- **Question**: Are the two boxed characters the same person?
- **Ground-Truth Answer**: No
- **Model**: Qwen2.5-VL-7B-Instruct
- **Predicted Answer**: Yes
- **Failure Analysis**: Lack of focus on cross-panel details. The model fails to capture subtle differences (e.g., hair colors) between characters when they appear in distant panels.
- **Full Predicted Answer**:

```
"<think>\nThe two boxed characters appear to be the same person. The character in the purple box is shown in
a close-up, and the character in the orange box is shown in a group scene. Despite the different poses and
the presence of other characters, the facial features and overall appearance are consistent, suggesting they
are the same individual.\n</think>\n\n<answer>\nYes\n</answer>"
```

**Panel Reordering**

- **Question**: There are two possible spots, A and B, where a provided missing panel could go. Based on the story flow, which spot makes more sense?
- **Ground-Truth Answer**: B
- **Model**: GPT-4o
- **Predicted Answer**: A
- **Failure Analysis**: Insufficient narrative construction. The model fails to build a coherent ordered storyline across candidate positions and neglects to leverage textual cues that could aid story reconstruction.
- **Full Predicted Answer**:

```
"<think>The missing panel shows people being well-fed and in good condition, with food being distributed from
large cauldrons. In the existing panels, there is a sequence where characters are led to a dining area and
eat abundantly. The missing panel logically fits before the scene where they are shown eating, as it sets up
the context of the food distribution. Therefore, the missing panel should go in spot A, as it introduces the
abundance of food before the characters are shown eating.</think> <answer>A</answer>"
```

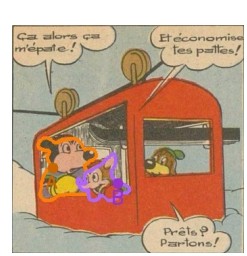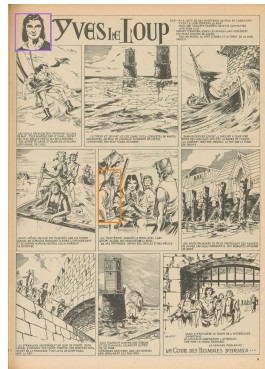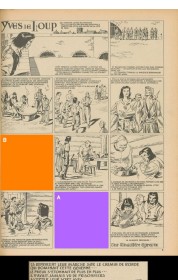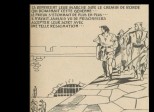

Figure 4: VQA instances of common failures: (1) Depth Comparison (2) Character Identification (3) Panel Reordering.

## E.2 MORE EXAMPLES OF AI4VA-FG

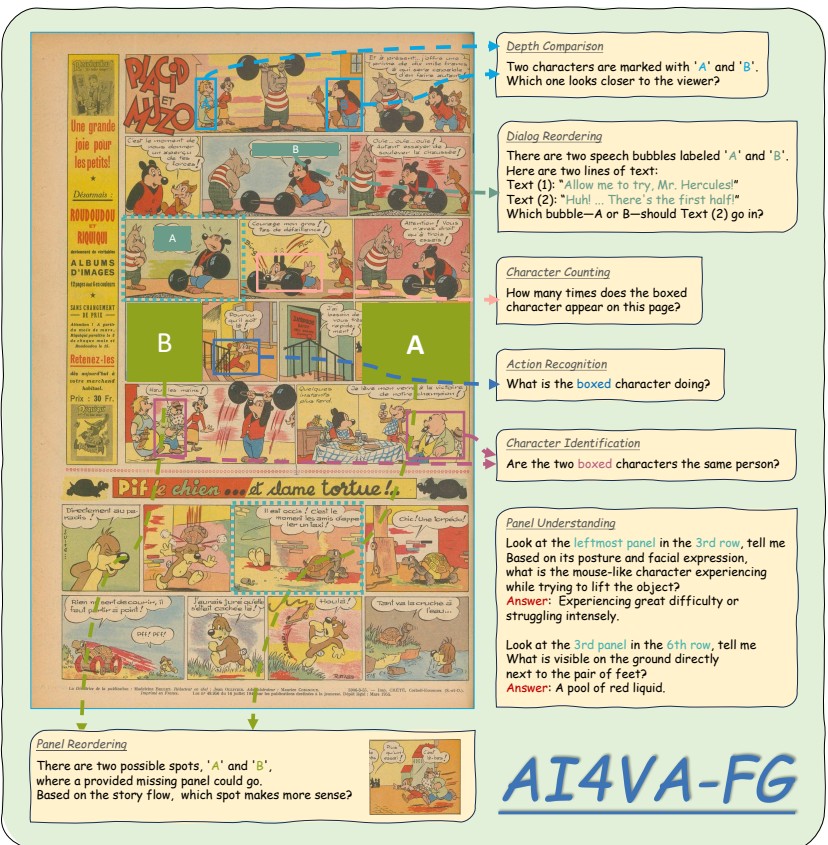

Figure 5: The full page image containing VQA instances shown in Fig.1.

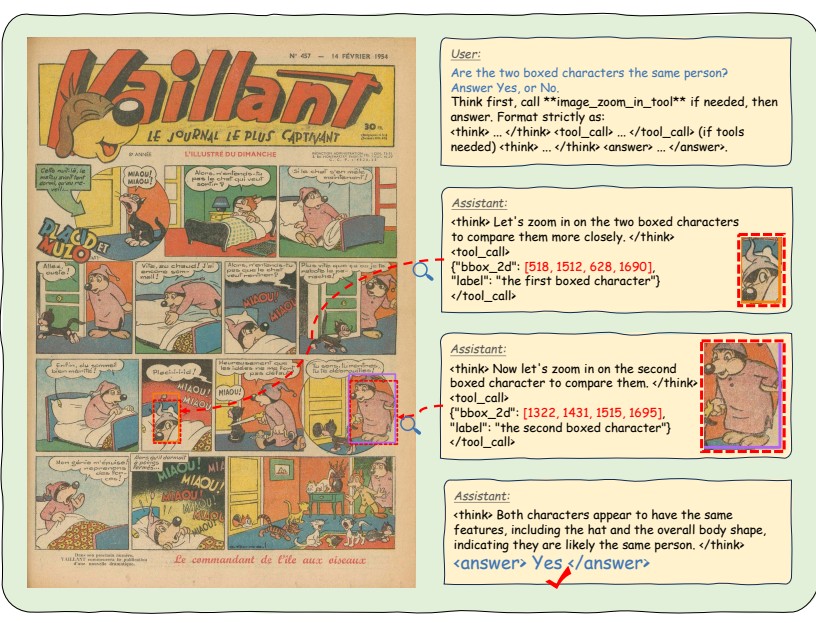

Figure 6: An example multi-turn conversation with zoom-in tool calling.

# F    EXPERIMENTS

## F.1    MODEL SELECTION

Our evaluation suite encompasses both proprietary and open-source MLLMs that represent the current state-of-the-art. For proprietary systems, given economic constraints, we select Gemini-2.5-pro (Team, 2025a) and GPT-5-mini (Hurst et al., 2024) as leading commercial baselines. On the open-source side, given computational constraints, we primarily focus on models under 10B parameters, including Qwen2.5-VL-3B/7B/32B (Team, 2025b), and InternVL3-8B/9B (Zhu et al., 2025). Most other open-source models on the VLM leaderboard share the same LLM or ViT backbones as these representatives, making our selection broadly representative of current open-source progress.

## F.2    EXPERIMENTAL SETUPS FOR FINETUNING

We train the open-ended and closed-ended tasks in Tab. 3.1 independently and report the results in Tab. 5.1. For closed-ended tasks, SFT is trained jointly across all tasks, while RL first uses sequential training across three categories followed by merged training, since starting with all tasks simultaneously causes convergence instability. Under sequential training, earlier tasks exhibit performance degradation as new categories are introduced, suggesting that the 7B base model, combined with the current dataset size, lacks sufficient capacity to generalize across all tasks concurrently. We also conduct category-wise training for both SFT and RL, which leads to greater improvements on most tasks, and confirm that the overall interpretation of results in Section 5 remains consistent.

Table 9: Hyper-Parameters for SFT Training

| Hyper-parameter | Value |
| --- | --- |
| Learning Rate | $1 \times 10^{-5}$ |
| Number of Epochs | 3 |
| Batch Size | 16 |
| Optimizer | AdamW |
| Learning Rate Scheduler | cosine |
| Warmup Ratio | 0.1 |
| Number of GPUs | 4 |

Table 10: Hyper-Parameters for RL (GRPO) Training

| Hyper-parameter | Value |
| --- | --- |
| Learning Rate | $1 \times 10^{-6}$ |
| Number of Steps | 200 |
| Rollout Batch Size | 16 |
| PPO Mini Batch Size | 16 |
| Num of Responses per Sample | 8 |
| Max Prompt Length | 10280 |
| Max Response Length | 4096 |
| Max Response Length (Region-Aware) | 4096 * 5 |
| KL Coefficient | 0.04 |
| Warmup Ratio | 0.0 |
| Rollout Engine | vLLM (0.8.2) |
| RL Engine | verl (0.2.0.dev0) |
| Number of GPUs | 8 |

### F.3 COMPARISON OF TOOL-USAGE BEHAVIOR DURING TRAINING ACROSS THREE REWARD STRATEGIES

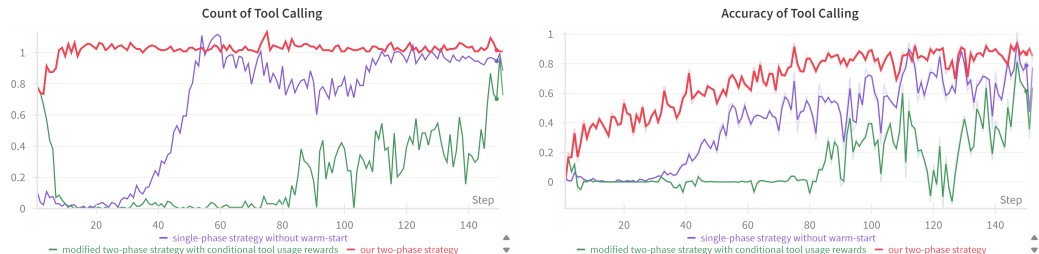

Figure 7: **Comparison of three reward strategie**s: (1) single-phase strategy without warm-start, (2) modified two-phase strategy with tool usage reward conditional on the final answer's correctness, and (3) our two-phase strategy (displaying only the second phase starting after 16 warm-start steps). Strategy (1) yields the most efficient tool learning.

## G PROMPT TEMPLATES

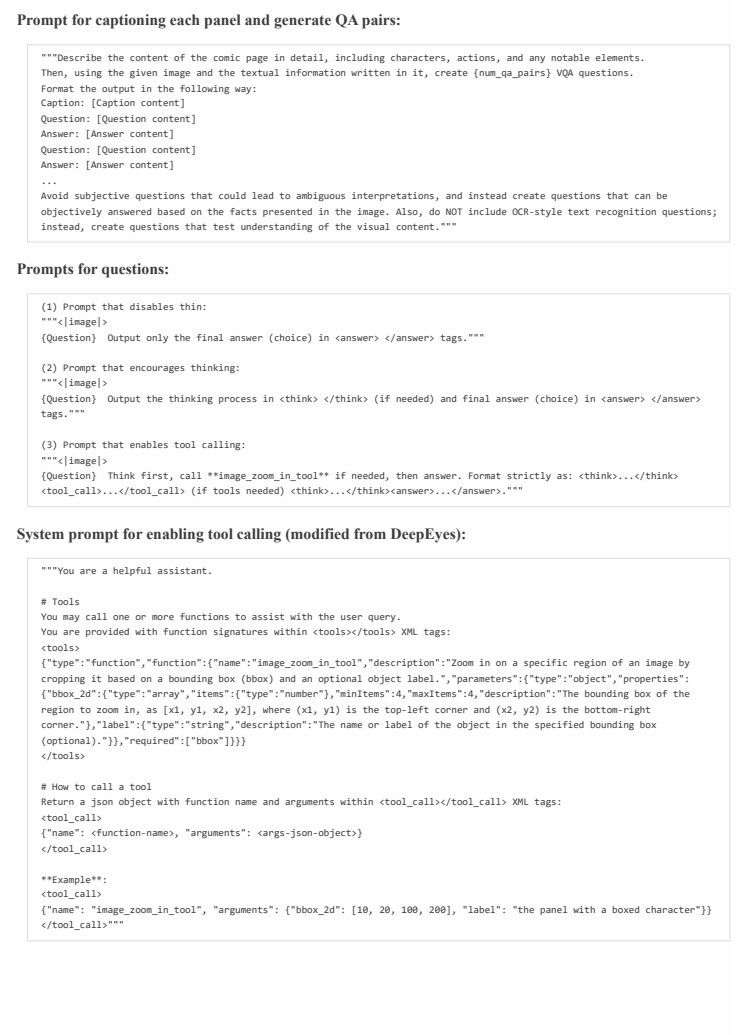