# OpenReview forum: "Zooming into Comics: Region-Aware RL Improves Fine-Grained Comic Understanding in Vision-Language Models"
_ICLR.cc/2026/Conference — Submitted to ICLR 2026_

### Official Review · Reviewer_sccJ · 2025-10-26

**Soundness:** 3
**Presentation:** 3
**Contribution:** 2
**Rating:** 4
**Confidence:** 4

**Summary:**

This paper tackles the poor fine-grained recognition and reasoning of VLMs on comics. The authors first introduce AI4VA-FG, a new benchmark to quantify this problem, identifying a key failure in SOTA models: a lack of selective attention. Inspired by this, the paper proposes Region-Aware Reinforcement Learning, a novel RL framework that trains the model to autonomously "zoom-in" on relevant regions. It uses a hierarchical reward function that jointly rewards final answer accuracy and the spatial accuracy (IoU) of the zoom-in action, leading to significant performance gains on fine-grained tasks.

**Strengths:**

1. The target issues of the paper are meaningful and worth exploring and the motivation is clear.
2. The paper is well written and easy to follow.
3. AI4VA-FG is a well-designed and challenging benchmark for fine-grained comic understanding, providing a strong analysis of SOTA model weaknesses.
4. The RARL reward function is novel and effective. It simultaneously rewards useful zooms (contributing to a correct answer) and accurate zooms (high IoU), making it a more sophisticated and effective strategy than vanilla RL.

**Weaknesses:**

1. The $R_{tool-acc}$ reward (Eq 3) requires ground-truth bounding boxes for the region of interest. This supervision is unavailable in most real-world VQA tasks, severely limiting the method's applicability to datasets without such dense annotations.
2. The method's success is tied to the zoom-in tool. It's unclear how this "region-aware" reward strategy would generalize to more complex agentic tasks requiring a wider variety of tools (e.g., crop, enhance, read_text).
3. The "Panel Understanding" task relies on LLM-as-a-Judge for evaluation. Furthermore, a footnote in Table 3 admits Gemini's high score is "not... a fair measure" as it generated the data, compromising the reliability of this task's results.
4. The method introduces a two-stage RL process with a complex, GT-dependent reward function. The complexity, stability, and computational cost of this training compared to simple SFT are not fully discussed.

**Questions:**

See weakness

---

### Official Review · Reviewer_Nz5D · 2025-10-30

**Soundness:** 2
**Presentation:** 2
**Contribution:** 2
**Rating:** 2
**Confidence:** 5

**Summary:**

This work first introduces AI4VA-FG, a fine-grained benchmark for evaluating VLM-based comic understanding. The benchmark spans tasks from foundational recognition and detection to high-level character reasoning and narrative construction. Meanwhile, they evaluated SOTA models on this benchmark, revealing significant deficits in their comic understanding capabilities. Furthermore, the authors systematically evaluated post-training methods, including SFT and RL, on improving model performance across comic understanding tasks. Then they propose Region-Aware Reinforcement Learning (RARL), a novel RL framework to enhance model performance on comic understanding tasks. When this strategy was applied to the Qwen2.5-VL model, it demonstrated significant improvements across the benchmark's tasks.

**Strengths:**

- The authors introduce a novel, fine-grained benchmark which demonstrates that existing state-of-the-art models have deficiencies in comic understanding tasks, highlighting a clear real-world application value. Detailed composition of the benchmark is also provided.
- Propose a Region-Aware Reinforcement Learning framework that effectively addresses the aforementioned comic understanding challenges. This framework learns where and when to zoom in a manner akin to how humans process complex visual information, and its effectiveness is proven by experimental results to some extent.
- Provide the categorization and definition of questions for the comic understanding task.

**Weaknesses:**

- It’s understandable that the authors only tested their RARL framework on Qwen2.5-VL models due to economic and computational constraints. However, this still needs more analysis and explanation. The paper should provide more detailed computational reports and visualizations of the training process to better substantiate the strategy's generality.
- The paper lacks a numerical analysis of the reward components during training (e.g., their fluctuation range or trends), and provides no corresponding visualizations. This omission makes it difficult for readers to deeply understand the specific role and impact of each reward component within the proposed strategy.
- The total reward is formulated as a direct summation of three components, which is a crude approach. Given that the paper's contribution hinges on the specially designed tool reward (R_{tool}), this component should have been explicitly weighted, and its impact analyzed more extensively.
- The tool reward function's design appears highly sensitive to the final answer's correctness. While the use of an indicator function to link tool usage to the final outcome is understandable, this raises concerns about the framework's stability. The paper fails to provide sufficient theoretical or numerical analysis to validate the stability of this reward mechanism.
- Furthermore, the paper provides no clear justification for why directly summing R_{tool-count} and R_{tool-acc} is a good paradigm. The analysis is unclear on how these two components interact during the training process or why this specific additive design was chosen.
- The ablation studies focus primarily on SFT and RL rather than on RARL, which deviates from the paper's main methodological contribution. The single ablation related to RARL, which is removing the constant coefficient in the reward equation, is insufficient to clearly analyze the contributions of the RARL framework's components. And the ablations seem to merely test some existing methods on the new benchmark rather than validate the proposed method.
- The calculation of R_{tool-acc} relies on IoU, which depends on the annotations of the target region. This design choice limits the framework's generality, suggesting that RARL may only be applicable to the authors' proposed benchmark or a small subset of tasks where such specific information is available.
- The case analysis is insufficient. Given that the paper previously proposed many classifications and definitions for comic understanding, it could provide more failure cases, as well as more zoom-in tool calling examples after applying RARL.
- The related work on zoom-in mechanisms feels incomplete. Given that "zoom-in" or selective attention is a natural cognitive process, its application in other computer vision tasks could be more thoroughly discussed.
- The experiments are insufficient to adequately validate the contributions proposed in the methodology.

**Questions:**

- Can the authors provide a supplementary theoretical or empirical justification for designing the reward strategy in the RARL framework this way? Why were the various rewards not weighted?
- Provide clearer visualization results and numerical analysis regarding the  RARL framework.
- Present more ablation study, hyper-parameter study and analysis on the proposed reward strategy.

---

### Official Review · Reviewer_xSTF · 2025-10-31

**Soundness:** 3
**Presentation:** 3
**Contribution:** 3
**Rating:** 4
**Confidence:** 3

**Summary:**

The paper “Zooming into Comics: Region-Aware RL Improves Fine-Grained Comic Understanding in Vision-Language Models” introduces AI4VA-FG, the first comprehensive benchmark for fine-grained comic understanding using vision-language models (VLMs). It includes seven tasks ranging from low-level recognition (e.g., action and depth detection) to high-level reasoning (e.g., dialogue and panel reordering), all with dense annotations. The authors systematically evaluate state-of-the-art proprietary and open-source VLMs, showing that even leading models like GPT-4o and Gemini-2.5 struggle with spatial perception, character tracking, and narrative reasoning. To address this, they propose Region-Aware Reinforcement Learning (RARL) — a training method that enables models to “zoom in” on relevant regions dynamically, improving both grounding and reasoning. Experiments demonstrate that RARL significantly enhances fine-grained recognition and storyline tasks, narrowing the gap between open and proprietary models.

**Strengths:**

The paper makes some contributions by addressing the overlooked challenge of comic understanding with a new fine-grained benchmark (AI4VA-FG) and a creative training approach, Region-Aware Reinforcement Learning (RARL).
Originality: It tackles an underexplored yet challenging domain of comic understanding by introducing a fine-grained benchmark (AI4VA-FG) that surpasses existing visual narrative datasets. The proposed Region-Aware Reinforcement Learning (RARL) framework is also an inventive adaptation of the “Thinking with Images” paradigm, demonstrating a novel approach to incorporating spatially grounded reasoning into VLMs.
Quality: The quality of the work is good, supported by thorough benchmarking, ablations, and comparisons across open and proprietary models.
Clarity: The paper is easy to understand, with clear figures and logical organization.
Significance: This work can be extended by adding more comic datasets and incorporating basic
tasks such as speaker identification. AI4VA-FG and RARL could shape future research on spatial reasoning and multimodal understanding beyond the comic domain.

**Weaknesses:**

1. I think the dataset construction pipeline is somehow missing in this paper. It does not show the details about how this dataset is constructed based on AI4VA dataset. I think this is one of the key part of this paper.
2. The style of the comic is very limited. It is only sourced from two mid-twentieth-century FrancoBelgian comics series. However, there are many other kinds of comics in the world. This is a huge limitation for the application of this dataset.
3. The experiment also shows that the cross-domain generalization ability on comics is poor on current VLMs. Therefore, this dataset may only benefit VLMs for this specific kind of comic.

**Questions:**

The pipeline of dataset construction should be explained in detail. So that other people can use this pipeline for annotating more comic datasets.

---

### Official Review · Reviewer_EDTe · 2025-11-01

**Soundness:** 3
**Presentation:** 3
**Contribution:** 3
**Rating:** 6
**Confidence:** 4

**Summary:**

This paper proposes a comic-centric benchmark AI4VA-FG. AI4VA-FG focuses on full-page, long-form comic content. It includes 7 tasks, encompassing panel understanding, depth comparison, action recognition, character counting, dialog reordering, character identification, and panel reordering. Although existing VLMs struggle in the comic domain, they propose a region-aware reinforcement learning method to address this challenge.

**Strengths:**

- The task in the comic domain is interesting and niche. They categorize 7 new sub-tasks, and each of them features some understanding capabilities. Existing models do not perform very well in this domain.
- They propose a region-aware reinforcement learning to incentivize models to zoom in on images when needed. It helps the performance significantly across different models.
- Multiple models are used in the experiments.

**Weaknesses:**

- Even though existing methods fall short in comic tasks, it is not very convincing why the comic task is important and can benefit other tasks in the community.
- They do not have much technical contribution. The SFT and RL are not novel concepts. While for comic books and small texts, it is natural to use zoom-in tools, it is the only tool that has a special reward in the training. It is doubtful how many cases in the test split are relevant to the zoom-in tool usage and if there are other important tools to use.

**Questions:**

- Other than the comic domain is under exploration, what else makes this task important? How can it benefit other, more general abilities of VLMs?
- Could you provide tags for the benchmark you propose? For each QA in the dataset, could you tag the tool that is recommended to use? Could you show the distribution, especially the zoom-in portion?

---

### Meta-Review · Area_Chair_gdEW · 2026-01-01

**Summary:**

In the initial reviews, three reviewers leaned toward rejection and one toward acceptance. As no rebuttal was provided, the consensus is that the paper does not meet the bar for ICLR.

**Reviewer Concerns:**

No rebuttal is provided.

**Reviewer Scores:**

6, 4, 4, 2

---

### Decision · Program_Chairs · 2026-01-26

Reject